# Surf smelt accelerate usage of endogenous energy reserves under climate change

Megan Russell[1]*, M. Brady Olson[1], Brooke A. Love[2]

**1** Department of Biology, College of Science and Engineering, Western Washington University, Bellingham, WA, United States of America, **2** Department of Environmental Sciences, College of the Environment, Western Washington University, Bellingham, WA, United States of America

\* meg83@hawaii.edu

**Data Availability Statement:** All data files will be available from the SEANOE database: Russell Megan (2018). Surf Smelt Embryo and Larvae Data. SEANOE. https://doi.org/10.17882/85830.

## Abstract

Surf smelt (*Hypomesus pretiosus*) are ecologically critical forage fish in the North Pacific ecosystem. As obligate beach spawners, surf smelt embryos are exposed to wide-ranging marine and terrestrial environmental conditions. Despite this fact, very few studies have assessed surf smelt tolerance to climate stressors. The purpose of this study was to examine the interactive effects of climate co-stressors ocean warming and acidification on the energy demands of embryonic and larval surf smelt. Surf smelt embryos and larvae were collected from spawning beaches and placed into treatment basins under three temperature treatments (13°C, 15°C, and 18°C) and two $p$CO$_2$ treatments (i.e. ocean acidification) of approximately 900 and 1900 µatm. Increased temperature significantly decreased yolk size in surf smelt embryos and larvae. Embryo yolk sacs in high temperature treatments were on average 7.3% smaller than embryo yolk sacs from ambient temperature water. Larval yolk and oil globules mirrored this trend. Larval yolk sacs in the high temperature treatment were 45.8% smaller and oil globules 31.9% smaller compared to larvae in ambient temperature. There was also a significant positive effect of acidification on embryo yolk size, indicating embryos used less maternally-provisioned energy under acidification scenarios. There was no significant effect of either temperature or acidification on embryo heartrates. These results indicate that near-future climate change scenarios may impact the energy demands of developing surf smelt, leading to potential effects on surf smelt fitness and contributing to variability in adult recruitment.

## Introduction

Surf smelt (*Hypomesus pretiosus*; Girard 1854 [1]) are small, schooling forage fish that provide a critical link in the marine food web by facilitating energy transfer from primary producers to higher trophic levels, including numerous species of sea birds, marine mammals, and other commercially important fish [2–5]. Surf smelt range extends from Prince William Sound, Alaska, to southern California [6], and they interact closely with the nearshore environment, especially during spawning events and juvenile development. Like other forage fish (e.g. herring, sand lance), surf smelt require unaltered shoreline to thrive [7, 8], making them

**Funding:** The project described in this publication was supported by the Department of the Interior Northwest Climate Adaptation Science Center (NW CASC- https://nwcasc.uw.edu/) through a Cooperative Agreement (G17AC000218) from the United States Geological Survey (USGS). Its contents are solely the responsibility of the authors and do not necessarily represent the views of the NW CASC or the USGS. This manuscript is submitted for publication with the understanding that the United States Government is authorized to reproduce and distribute reprints for Governmental purposes. The funders had no role in study design, data collection and analysis, decision to publish, or preparation of the manuscript. Award #:33491

**Competing interests:** The authors have declared that no competing interests exist.

indicators of overall marine system health. Any reductions to their population abundance can have cascading effects on marine ecosystem productivity and functioning [4, 5]. Despite their ecological importance, they are an understudied forage fish in the North Pacific ecosystem.

Surf smelt spawn on gravel beaches during high tides, utilizing the upper third of a beach's tidal range, typically at tidal heights of seven feet or above [4, 7, 9]. Female surf smelt deposit eggs which adhere to gravel substrate [4, 8]. Once deposited and fertilized, eggs and their developing embryos are subjected to daily exposure in both air and seawater throughout their maturation. Consequently, embryos experience a dynamic range of environmental conditions over diel cycles [8, 10]. During this critical time, surf smelt embryos develop by utilizing maternally provided (endogenous) energy in the form of yolk and lipid droplets for approximately 14 to 21 days depending on ambient temperature and sediment humidity [4, 10]. When embryos are fully developed, hatching is triggered by physical disturbance, primarily resulting from tidal activity and wave action [4, 7, 10]. Once hatched, the planktonic surf smelt larvae continue to utilize yolk reserves and lipids contained within an oil globule for energy and development [4] until exogenous feeding begins. Any accelerated usage of stored energy in forage fish could negatively affect performance and survival, and ultimately, adult recruitment [11].

One mechanism that may force accelerated usage of endogenous energy reserves in surf smelt embryos and larvae is energy demand for compensatory processes like maintaining physiological homeostasis under temperature stress [12, 13]. For an obligate beach spawner like surf smelt, this is reason for concern. Shoreline development is increasingly removing nearshore vegetation and the shading it provides for developing surf smelt embryos, causing a significant increase in beach gravel temperature and concurrent decreases in gravel humidity [7]. A survey of modified and natural beaches in the Salish Sea found that temperatures on modified beaches were, on average, 4.7˚C higher than temperatures of natural beaches [7]. Research on the effects of temperature on surf smelt have predominantly focused on the effects of air temperature and desiccation stress on mortality and hatching in the field [7, 10]. For example, increased air temperature was linked to increased developmental rates in surf smelt embryos [7], indirectly suggesting increased yolk utilization.

The effects of temperature on larval fish fitness are often amplified under coincident environmental stressors [e.g. 14–16]. One potential concomitant stressor is ocean acidification (OA). OA is a suite of chemical changes in marine carbonate parameters (e.g. pH, $p\mathrm{CO_2}$, $[\mathrm{CO_3}^2]$) that are driven on a global scale by the rise in total seawater carbon ($C_T$) through dissolution of atmospheric $\mathrm{CO_2}$ into seawater. Exposure of embryonic and larval fish to both temperature stress and OA conditions may drive reallocation of energy reserves. OA is shown to have wide-ranging effects on marine organisms across multiple taxonomic groups [17], including on related species of forage fish [18–20]. Temperate cold-water ecosystems like the northeast Pacific Ocean are vulnerable to OA due to the ability of cold water to absorb atmospheric gasses more efficiently than warm water [21]. Additionally, seasonal upwelling and strong tidal mixing introduce deep waters enriched in $\mathrm{CO_2}$ into surface waters [22, 23], acting to exacerbate global OA processes and signals. In the Salish Sea's urbanized Puget Sound estuary, where surf smelt spawning is widespread [8], observed minimum pH in surface waters varied between 7.4 and 7.7 [22], indicating $p\mathrm{CO_2}$ values on the order of 1000–2000 μatm. Global ocean pH values are predicted to reach about 7.7 at the end of the century under a business as usual scenario (RCP8.5) [24]. Due to the intertidal spawning behavior of surf smelt, which in some locations across the northeast Pacific Ocean occurs year-round [4], embryos are not continuously submerged during embryogenesis. Thus, the high daily and seasonal variability of $p\mathrm{CO_2}$ and pH exposure the embryos experience may select for embryo phenotypes that are robust to variable and at times high $p\mathrm{CO_2}$. However, the pelagic lecithotrophic larvae

remain nearshore, where $pCO_2$ concentrations in northeast Pacific Ocean ecosystems can be extraordinarily high, and beyond those even predicted for global averages at the end of this century [22]. As such, surf smelt larvae under OA may consume yolk and the oil globule at high and, if an interaction between temperature and $pCO_2$ exists, at exacerbated rates.

The purpose of this study was to investigate the combined effects of elevated seawater temperature and OA on the energy demands of surf smelt embryos and larvae. This was accomplished through measuring yolk exhaustion and heart rates in embryos, and yolk and oil globule exhaustion in surf smelt larvae, under three temperatures and two $pCO_2$ conditions. We hypothesized that energy demand, measured as embryo heart rate and yolk sac/oil globule exhaustion, for both embryos and larvae would increase under elevated seawater temperature and $pCO_2$ (i.e. acidification) as isolated stressors. We also predicted that the highest energy usage would be observed under simultaneous increased temperature and elevated $pCO_2$ due to the additive effect of these climate stressors on compensatory homeostatic processes.

## Methodology

### Experimental set-up

All experimental seawater was collected from the Salish Sea at the Shannon Point Marine Center (SPMC) in Washington State, U.S.A. This water was dispensed into a header tank that gravity fed seawater into six mixing basins. In each mixing basin, submersible powerhead pumps (Marineland® Maxi-jet 900) circulated water using magnetically driven impellers. Three mixing basins continuously received a slow feed of pure $CO_2$ gas from an 8 channel Masterflex® L/S Digital Drive peristaltic pump attached to a 20-lb food grade $CO_2$ gas cylinder. This flow of pure $CO_2$ (19 mL/min), tuned to achieve the desired $pCO_2$ given the flux of water into the tanks, was fed into the intakes of the powerhead pumps and completely dissolved while passing through the impeller chamber and associated turbulence [25]. This method of $CO_2$ amendment is suitable for creating large volumes of water with elevated $pCO_2$, as it does not depend on slower gas exchange processes. These pure $CO_2$ additions generated two $pCO_2$ treatments, ambient and elevated. Each mixing basin delivered water to three treatment basins held at the different temperatures (ambient seawater ~13˚C, 15˚C, and 18˚C).

The tanks were heated using 100W Aqueon® submersible heaters attached to Inkbird® temperature regulators. The temperature regulators engaged the heaters anytime the temperature deviated below 0.3˚C of the set temperature. Treatment temperatures were chosen to represent ambient seawater temperatures during the summer surf smelt spawning period in the Salish Sea (13˚C), predicted seawater temperatures in the Salish Sea by the year 2040 (15˚C; [16]), and the current average sediment temperature (18˚C) observed at Salish Sea surf smelt spawning beaches that have been modified by human activity [7]. The two $pCO_2$ conditions generated in the mixing tanks were modified slightly in the treatment basins as a result of temperature driven differences in solubility and in $pCO_2$, resulting in a range of OA scenarios. $pCO_2$ across temperature treatments in ambient basins was about 750–1000 µ atm and about 1700–2200 µatm in elevated basins, with a $pH_{NBS}$ range of approximately 7.42–7.88. No $CO_2$ additions were made to the ambient seawater treatments, which were representative of the ambient seawater at the SPMC. $CO_2$ additions for the elevated treatment were chosen to result in $pCO_2$ similar to those currently observed in the Salish Sea during naturally occurring $CO_2$-enrichment [22], and elevated future levels that may result when global anthropogenic OA combines with natural regional acidification events. Treatment basins were covered with 5/8" acrylic sheets to limit gas exchange during experiments.

## Collection

Surf smelt eggs containing embryos and their attached sediment were collected from Fidalgo Bay, a shallow embayment located in the Salish Sea's northern Puget Sound in Washington State, U.S.A (48.4833816,-122.5868376). Embryos were transported to the SPMC where they were examined via stereomicroscopy to determine age. Experiments began if the embryos were determined to be < 24 hours old. Age was determined using the guidelines of Moulton and Penttila [26]. After aging, embryos were dispensed into experimental treatment basins. Approximately 50 embryos were added to each of 72, 200-mL glass bowls. These bowls were divided evenly into 18 treatment basins (4 bowls per basin). Our experimental design resulted in embryos being continuously bathed by seawater and does not replicate the sporadic inundation surf smelt embryos experience in nature. Nonetheless, our embryonic results are informative of the physiological responses of surf smelt embryos in response to abiotic stress and can be used to inform other studies.

If the collected embryos were near hatch, then a larval experiment was started. For these experiments, the embryos and attached sediment were placed into 12-L bins where they were gently oscillated for 1-minute increments to simulate wave disturbance and initiate hatching. Hatched larvae were collected, and the bins were oscillated again until no additional hatch was observed. Approximately 15 freshly hatched larvae were distributed into each of 72, 200-mL glass bowls which were divided evenly into 18 treatment basins (4 bowls per basin).

## Water chemistry measurements

Temperature measurements from each basin were taken every 30 minutes by Onset HOBO® data loggers to verify that temperature remained consistent throughout the experiments. Daily pH measurements from treatment water within basins were taken using a hand-held Orion Star A329 pH conductivity meter, calibrated with NBS-buffers prior to use each day. Spectrophotometric pH measurements were also taken, but the data were compromised and could not be used. While it is clear that pH measured with a glass electrode is not preferred, an extensive probe specific cross-comparison with pH values on the total scale derived from spectrophotometric pH methods ($pH_T$) was carried out, yielding a strong linear relationship relating $pH_{NBS}$ to $pH_T$ ($r^2$ 0.997). Seawater samples stored in 20-mL plastic scintillation vials were also collected three times per week throughout the experiment for $C_T$ analyses. Water samples for $C_T$ analysis were poisoned with 20 μL mercuric chloride ($HgCl_2$) to arrest microbial metabolism and stored for later analysis. Samples were analyzed for $C_T$ using an Apollo SciTech AS-C3. Sample salinity was measured using a refractometer, and salinity values and temperature were used to convert $C_T$ measurements from μmol/L to μmol/kg. Measurements of $C_T$ were calibrated against a standard curve created from reference material (CRM, Batch 179, Dickson, Scripps Institute of Oceanography). CO2SYS [27] was used to calculate all remaining components of the carbonate system including $pCO_2$, using K1 and K2 equilibrium constants refit by Millero et al. [28] and the sulfate dissociation constant by Dickson [29].

## Energy measurements

Surf smelt embryo yolk usage was measured daily over a 13-day experiment, with the exception of the ambient temperature treatment (~13°C). This treatment ended after 10 days due to natural warming of the ambient seawater which made it redundant with the 15°C treatment. Each day, 3 embryos from each of the bowls within each treatment basin were haphazardly selected and placed into 6-well plates labeled with its corresponding treatments. Embryos were photographed using a Leica M125 stereoscope attached to a Leica MC170 camera networked to Leica Suite software. After being photographed, embryos were removed from the

experiment to avoid repeated measurements. Photos were later analyzed using the software ImageJ [30] to determine the egg and total yolk area. These measurements were used to calculate the ratio of yolk area to total egg area. This ratio was used to account for initial differences in egg size.

Surf smelt embryo heart rate measurements were taken as another measure of energy demand. Due to heart rate being affected by fish developmental stage [31], heart rate measurements were taken within treatments at a specific, identifiable developmental stage: when the eye spots of the developing embryos darken [26]. When embryos were removed from glass bowls to be photographed for yolk measurements, if the eye spots had darkened since the previous day, the embryos were also video recorded for 10 seconds. Heart beats were later counted in each video.

Energy consumption of surf smelt larvae was assessed through a 3-day experiment, during which 2 larvae were haphazardly selected each day from each bowl within each treatment basin. Larvae were anesthetized using tricaine following the methods of Massee et al. [32]. This method limits larval movement and allows still images of the larvae to be taken. Leica software and a stereomicroscope were used to photograph the anesthetized larvae. After being photographed, larvae were removed from the experiment to avoid repeated measurements. Photographs were later analyzed using ImageJ [30] to determine yolk area and oil globule area.

## Statistical analysis

Larval yolk sac, oil globule size, and embryo heart rates were analyzed using linear mixed effect models with day, $pCO_2$, temperature, and their interactions as fixed factors, and experimental basin as a random factor. The ratio of yolk size to embryo size was analyzed using a generalized linear mixed effect model (family = binomial, link = logistic) with day, $pCO_2$, temperature, and their interactions as fixed factors, and experimental basin as a random factor. For each response variable measured, models were built with and without the interaction terms based on a combination of p-value significance ($\alpha = 0.05$) and AIC comparisons to choose the most appropriate model. Interaction terms that were not significant and did not improve the model fit were left out. Random slope or intercept terms were applied to experimental basin depending on model fit determined by AIC comparisons. Data from elevated and ambient $pCO_2$ treatments are configured categorically in figures for clarity, but $pCO_2$ was treated as a continuous variable in statistical models. All statistics were completed using R [33].

## Ethics approval

All experimental procedures were carried out in accordance with the recommendations and approval of the Animal Care and Use Committee at Western Washington University (protocol approval 19–001), and embryos were collected in collaboration with the Washington State Department of Fish and Wildlife.

## Results

### Treatment conditions

Direct injection of pure $CO_2$ into the elevated $pCO_2$ mixing tanks followed by equilibration of both ambient and elevated $pCO_2$ treatments to three temperatures resulted in two distinct $pCO_2$/pH conditions, modified somewhat by temperature (Table 1 embryo and larval yolk and oil globule experiments) (S1 Table heartbeat experiment). The average pH variability was similar between ambient and high $pCO_2$ treatments at comparable temperatures ($\pm$ 0.03 pH units) (Table 1). The $CO_2$ amendment in the elevated $pCO_2$ treatment resulted in an average increase in 42 μmol kg/SW relative to the ambient treatment (Table 1). Because $pCO_2$ is derived from

**Table 1. Treatment conditions.**

| Experiment | Conditions | *In-Situ* Measurements | | Discrete Samples | | |
| | | pH | Temperature | $p$CO$_2$ | $C_T$ | Salinity |
| | ($p$CO$_2$ +˚C) | (NBS Scale) | (˚C) | (µatm) | (µmol kg$^*$SW$^{-1}$) | |
| Embryo Yolk | ambient +13 | 7.88 ± 0.01 (36) | 13.77 ± 0.24 (36) | 793. 85 ± 28.59 (15) | 2015.06 ± 10.44 (15) | 30.8 ± 0.6 (15) |
| | ambient +15 | 7.87 ± 0.03 (42) | 14.89 ± 0.03 (42) | 749.37 ± 141.69 (15) | 2013.21 ± 16.53 (15) | 30.7 ± 0.5 (15) |
| | ambient +18 | 7.83 ± 0.02 (39) | 17.94 ± 0.23 (39) | 872.25 ± 133.73 (15) | 2010.92 ± 14.86 (15) | 30.6 ± 0.6 (15) |
| | elevated+13 | 7.56 ± 0.09 (39) | 13.44 ± 0.24 (39) | 1695.41 ± 337.05 (15) | 2049.61 ± 11.56 (15) | 31.1 ± 0.9 (15) |
| | elevated+15 | 7.56 ± 0.06 (39) | 15.02 ± 0.03 (39) | 1617.05 ± 324.02 (15) | 2050.43 ± 15.9 (15) | 30.9 ± 0.9 (15) |
| | elevated+18 | 7.53± 0.03 (39) | 17.93 ± 0.23 (39) | 1695.41 ± 368.98 (15) | 2047.64 ± 17.34 (15) | 30.6 ± 0.6 (15) |
| Larvae Yolk and Oil Globule | ambient +13 | 7.85 ± 0.01 (9) | 13.22 ± 0.35 (9) | 828.75 ± 25.25 (6) | 2032.91 ± 11.06 (6) | 31.0 ± 0.6 (6) |
| | ambient +14 | 7.83 ± 0.01 (9) | 14.15 ± 0.08 (9) | 888.25 ± 25.49 (6) | 2028.65 ± 12.21 (6) | 30.8 ± 0.4 (6) |
| | ambient +18 | 7.79 ± 0.01 (9) | 17.42 ± 0.15 (9) | 992.22 ± 39.38 (6) | 2022.67 ± 14.04 (6) | 30.6 ± 0.6 (6) |
| | elevated+13 | 7.54 ± 0.07 (9) | 13.43 ± 0.43 (9) | 1759.74 ± 316.37 (6) | 2067.31 ± 16.84 (6) | 31.2 ± 0.7 (6) |
| | elevated+14 | 7.51 ± 0.06 (9) | 14.15 ± 0.23 (9) | 1787.97 ± 247.35 (6) | 2077.95 ± 17.39 (6) | 30.9 ± 0.6 (6) |
| | elevated+18 | 7.51 ± 0.06 (9) | 17.32 ± 0.11 (9) | 1907.61 ± 309.72 (6) | 2065.77 ± 16.79 (6) | 31.0 ± 0.6 (6) |

Average seawater temperature, pH, $p$CO$_2$, and $C_T$ for two experiments. Data for the embryo heartrate experiment are presented in S1 Table. Data are shown as time-averaged means ± 1 SD of (n) measurements. pH and temperature were measured daily, while $C_T$ was measured 3 times per week. $p$CO$_2$ was derived from temperature, pH, and $C_T$ measurements.

temperature, pH, and $C_T$, the variability in all three measurements results in relatively high variability in $p$CO$_2$ estimates. The ambient $p$CO$_2$ tanks had an average standard deviation of ± 50.58 µatm $p$CO$_2$, while the elevated $p$CO$_2$ tanks had considerably higher variability, with an average standard deviation of ± 275.89 µatm (Table 1). Differences in mean $p$CO$_2$ between temperature levels within $p$CO$_2$ treatments and were generally less than 150 µatm, considerably smaller than the approximately 800 µatm difference between the ambient and elevated $p$CO$_2$ treatments (Table 1). Therefore, data are presented in the elevated $p$CO$_2$ and ambient $p$CO$_2$ treatments in figures for embryo and larval characteristics. Given the need to use electrode pH measurements, we acknowledge the limitation of these estimates of $p$CO$_2$.

## Embryo yolk usage and heartrate

The proportion of embryo yolk size relative to total egg size (Y:E) decreased throughout the duration of the experiment in all treatments with day having a significant negative effect on Y:E (Table 2; Fig 1). Temperature also had a significant, negative effect on Y:E throughout the 13-day experiment (Table 2; Fig 1). The average Y:E decreased by 21.6% after 13 days of incubation at 13˚C compared to a 28.2% decrease at 18˚C. This equates to Y:E being 7.3% smaller at 18˚C compared to 13˚C at the final time point. There was also a significant, positive effect of

**Table 2. Generalized linear model results for Y:E.**

| | *Value* | *Std. Error* | *z-value* | *p-value* |
| --- | --- | --- | --- | --- |
| *Intercept* | 6.5614 | 0.5062 | 12.960 | <2.00e-16 |
| *Day* | -0.49 | 0.0181 | -27.927 | < 2.00e-16 |
| *pCO2* | 0.4460 | 0.1024 | 4.356 | 1.32e-05 |
| *Temperature* | -0.2321 | 0.0303 | -7.654 | 1.94e-14 |

Summary of results of a generalized linear mixed effect model (family = binomial, link = logistic) examining the effect of day, temperature, and $p$CO$_2$ on embryo yolk sac relative to total egg size.

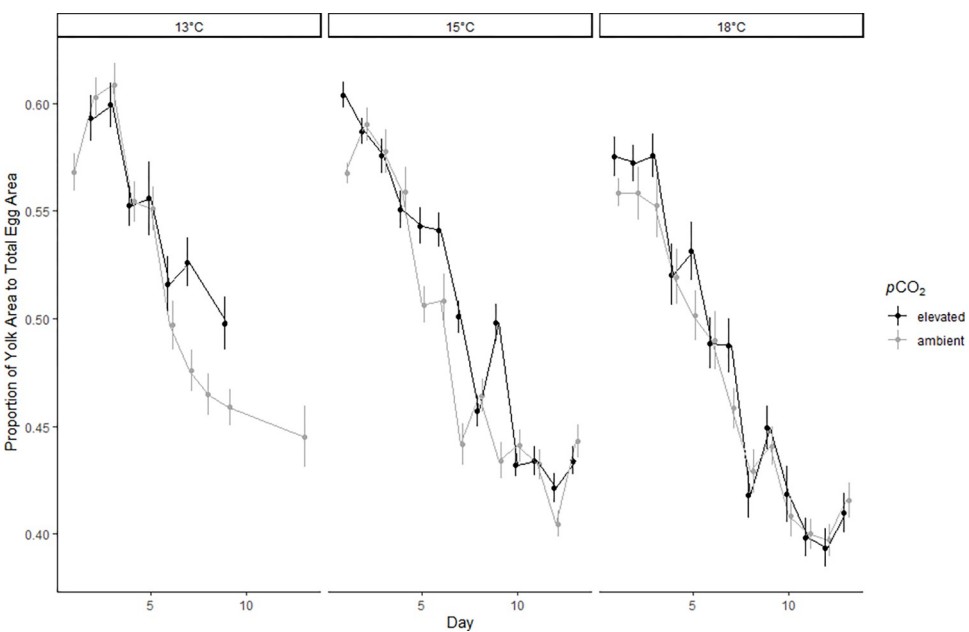

**Fig 1. Embryo yolk usage.** Proportion of yolk area to total egg area (Y:E) of surf smelt embryos (n = 216/day) that were submerged in treatment water for 13 days. Each point is an average measurement representative of the temperature treatment, $pCO_2$ treatment (points offset for visualization), and day with standard error shown as whiskers. Day, $pCO_2$ and Temperature were significant factors predicting embryo yolk usage.

elevated $pCO_2$ on Y:E (Table 2). Embryo heart rates were not significantly affected by temperature or $pCO_2$ stressors (Table 3). Surf smelt embryo heart rates in ambient temperature treatments were similar between $pCO_2$ treatments. As temperature increased, the average heartrate increased between 13°C and 18°C at elevated $pCO_2$ and a decreased at ambient $pCO_2$ but variability was high, and these patterns were not significant (S1 Fig).

## Larvae yolk and oil globule usage

Larval yolk size decreased over time in all treatments with day having a significant negative effect on larval yolk size (Table 3; Fig 2). Temperature also had a significant, negative effect on

**Table 3. Linear mixed effect model results.**

| | | Value | Std. Error | df | t-value | p-value |
|---|---|---|---|---|---|---|
| Embryo heart rate | *Intercept* | 11.8181 | 0.9540 | 65 | 12.3879 | <0.0000 |
| | *Temperature* | -0.0252 | 0.0230 | 26 | -1.0925 | 0.2846 |
| | *$pCO_2$* | 0.0008 | 0.0005 | 65 | 1.8712 | 0.0657 |
| Larval yolk sac size | *Intercept* | 0.3389 | 0.0270 | 383 | 12.5504 | <0.0000 |
| | *Day* | -0.0283 | 0.0019 | 383 | -14.3675 | <0.0000 |
| | *$pCO_2$* | 0.000004 | <0.0000 | 16 | 0.6234 | 0.5418 |
| | *Temperature* | -0.0111 | 0.0017 | 383 | -6.4106 | <0.0000 |
| Larval oil globule size | *Intercept* | 0.0823 | 0.0071 | 383 | 11.5417 | <0.0000 |
| | *Day* | -0.0075 | 0.0007 | 383 | -10.8844 | <0.0000 |
| | *$pCO_2$* | -0.000001 | <0.0000 | 16 | -0.8062 | 0.4319 |
| | *Temperature* | -0.0018 | 0.0005 | 383 | -4.1379 | <0.0000 |

Summary of linear mixed effect models examining the effect of day, temperature, and $pCO_2$ treatment level on multiple parameter measurements.

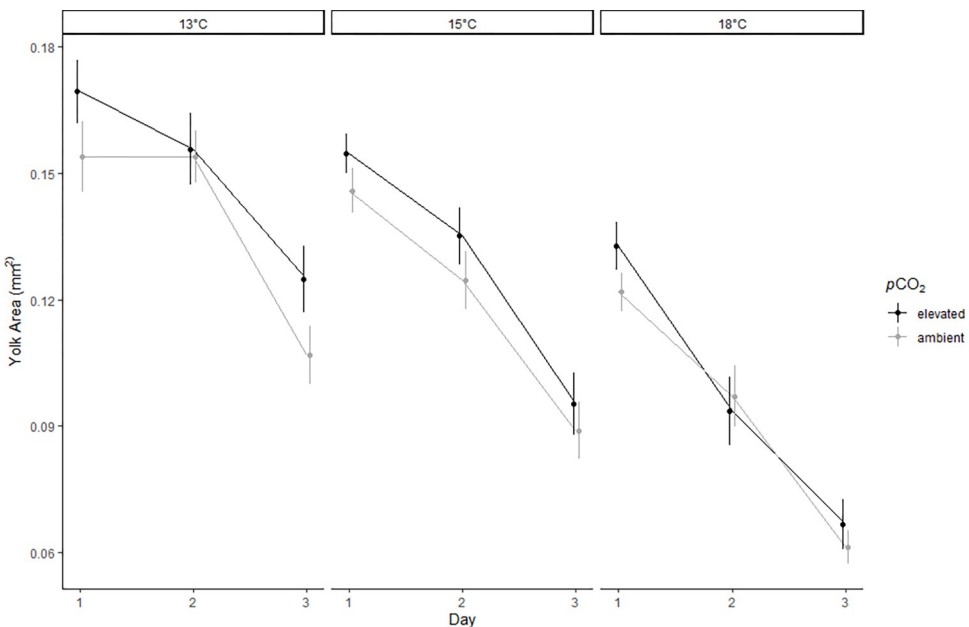

**Fig 2. Larvae yolk usage.** Yolk sac area of surf smelt larvae (n = 144/ day) that were submerged in treatment water for 3 days. Data are averages representative of the temperature treatment, $pCO_2$ treatment (points offset for visualization), and day with standard error shown as whiskers. Day, and Temperature were significant factors predicting larvae yolk usage.

larval yolk size throughout the duration of the experiment (Table 3; Fig 2) and for every 1˚C increase in temperature, the average larval yolk size decreased by 0.011 mm$^2$ (Table 2). The average size of larval yolk sacs decreased by 27.1% at 13˚C and 49.8% at 18˚C over the duration of the experiment. Larval yolk size was not significantly affected by $pCO_2$ (Table 3; Fig 2).

Surf smelt oil globule size decreased over time in all treatments with day having a significant negative effect on oil globule size (Table 3; Fig 3). Temperature had a significant negative effect on the size of larval oil globules and for every 1˚C increase in temperature, the average oil globule area decreased by 0.0018 mm$^2$ (Table 3; Fig 3). Over the three-day incubation, oil globule size decreased by 25.7% at 13˚C compared to 43.2% at 18˚C. On average, oil globules at 18˚C were 31.9% smaller than oil globules at 13˚C on the final day of data collection (Fig 3). $pCO_2$ had no significant effect on oil globule size (Table 3).

## Discussion

Increased energy usage due to elevated metabolism across fish life histories can result in trade-offs in growth, performance, and reproduction [34–36]. We found that elevated temperature was a primary abiotic driver affecting early life history stages of surf smelt, accelerating yolk usage in both surf smelt embryos and larvae, and oil globule usage in larvae. Our results showing increased usage of endogenous energy in response to increasing temperatures agree with findings from numerous other fish species [e.g. 37–43], including the forage fish *Engraulis japonica* [38], *Clupea pallasi* [40] and *Clupea harengus* [41], *Brevoortia tyrannus* [42], and *Sardinops sagax* [43]. However, our results also suggest that surf smelt may be more tolerant to temperature variance than other fish species. Heming and Buddington [37] summarized the findings of 23 species of fish and showed that $Q_{10}$ values for yolk absorption in fish over a temperature span of 1–30˚C averaged 2.9. In this study we observed $Q_{10}$ values of 1.58 and 2.09 for surf smelt embryos and larvae, respectively. In the broadcast spawning Pacific sardine, Lasker

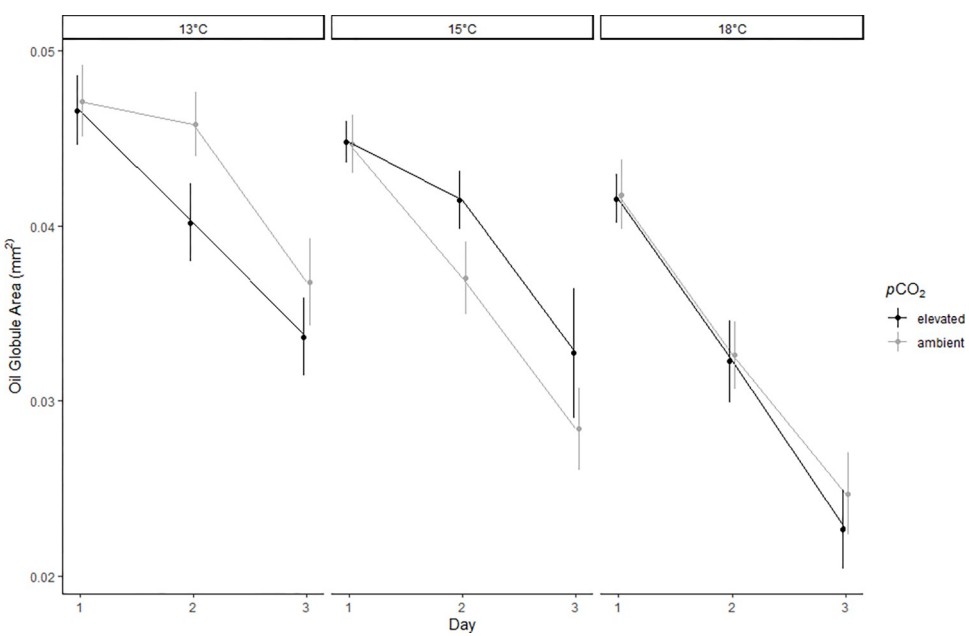

**Fig 3. Larvae oil globule usage.** Oil globule area of surf smelt larvae (n = 144/day) that were submerged in treatment water for 3 days. Each point is an average measurement representative of the temperature treatment, $pCO_2$ treatment (points offset for visualization), and day with standard error shown as whiskers. Day, and Temperature were significant factors predicting larvae oil globule usage.

[42] found a $Q_{10}$ of 4 for yolk absorption between environmental temperature range of 15 to 21°C. The fact that our observed $Q_{10}$ values are slightly below the averages summarized in Heming and Buddington [37] and well-below those reported in Lasker [43] suggests that surf smelt, while not immune to temperature effects, are less sensitive to temperature stress compared to other fish species. More robust tolerance to elevated temperature compared to other fish species may be an adaptation to withstand the wide-ranging temperatures surf smelt embryos experience during their high intertidal incubation, which can last between 10 and 21 days [4].

Elevated temperature was associated with a significant decrease in oil globule size in addition to decreased larval yolk area in this study, indicating ongoing taxation on endogenous energy in the larval stage. While fish embryos use free amino acids found in their yolk as their primary energy source prior to hatching, they switch to using fatty acids from their oil globule post hatching [44]. Ehrlich and Muszynski [45] proposed that in addition to being a source of energy for metabolism, the lipids that make up the oil globules in surf smelt likely aid in larval buoyancy. This was supported by findings in Blaxter and Ehrlich [46], who observed that Atlantic herring and European plaice were oriented head down in the water column post yolk absorption. These larvae were unable to feed even when offered food due to their heads sinking when attempting to follow prey. The rapid use of the oil globule in surf smelt at higher temperatures could result in compromised fitness when larvae enter the plankton, and buoyancy effects likely reduce exogenous energy acquisition through decreased swimming performance and prey capture efficiency. Swimming is arguably the largest sink for energy in fish larva [46], and an essential behavior for acquiring prey and for locating abiotic conditions that maximize fitness and survival. Thus, any environmental factor that accelerates the use of this endogenous energy source prior to the onset of feeding may affect larval survival and, ultimately, adult recruitment.

In the Salish Sea's Puget Sound, elevated temperatures can be driven by human activities. Regionally, approximately one third of the coastline has been modified by humans [47]. Human activities and development alter surf smelt spawning grounds by reducing nearshore vegetation and the shading it provides, causing a significant increase in beach gravel temperature and concurrent decreases in gravel humidity [7]. A survey of modified and natural beaches in the Puget Sound found that modified beaches had an average substrate temperature of 18.8°C, which was 4.7°C higher than the average temperature of natural beaches, and a temperature known to cause increased surf smelt embryo mortality [7]. The observed temperature increase caused by shoreline modification was represented by the high temperature treatment (18°C) in this study. It is likely that contemporaneous shoreline modifications and subsequent temperature effects are realigning surf smelt energy budgets and, potentially, the timing of hatching [7, 10]. Ongoing shoreline development coupled with the large proportion of already developed beaches in the Salish Sea poses an immediate threat as surf smelt rely on a relatively small proportion of beaches for spawning each year, and disturbance to any of them could have population-level implications [8]. Additionally, climate models predict that Salish Sea seawater temperatures will increase by approximately 1.2°C by the year 2040 [48]. This temperature increase was captured in this study as the 15°C treatment and was found to reduce yolk size by 22% compared to ambient seawater. Thus, findings here suggest that global climate change will accelerate surf smelt endogenous energy usage in the northeast Pacific Ocean.

We observed little direct effects of elevated $p\text{CO}_2$ on early life history stages of surf smelt, which is in agreement with recent findings on another Salish Sea forage fish, *Clupea pallasii* [40]. Embryo heartrate, larval yolk size, and oil globule size were not directly affected by variation in $p\text{CO}_2$, despite the very high concentrations used in this study. Only surf smelt embryos were directly affected by elevated $p\text{CO}_2$, with a significant positive effect on Y:E. A review of the effects of OA on marine fish metabolic rates found that 21% of studies reported a consistent reduction in metabolic rate in response to OA [49]. The positive effect of elevated $p\text{CO}_2$ on surf smelt embryos and the absence of a $p\text{CO}_2$ treatment effect on the surf smelt larval metrics may reflect the unique spawning behavior and habitat use of surf smelt, and an overall robustness to OA sensitivity. In comparison to open-ocean species, marine fish that spawn in the intertidal or near-shore habitat possess broad tolerance to environmental variability due to the high frequency and magnitude of environmental change [7, 40, 50, 51]. With respect to habitat-dependent $\text{CO}_2$ sensitivity, Baumann [52] called this the Ocean Variability Hypothesis. The surf smelt embryos used here were collected from a shallow embayment that experiences highly variable and ephemeral pH, ranging from $\leq 6.5$ to $> 8.5$, with the majority of observations between 7.15 to 8.45 [53]. Thus, the lack of a strong OA effect on surf smelt, even on early life histories, can be expected.

While this study shows that the utilization of endogenous energy reserves in early life-history stages of surf smelt are affected by elevated temperature and, to a lesser degree, $p\text{CO}_2$, it remains unknown whether these effects will be detrimental to later life history stages, or in annual population abundance and recruitment. Despite this limitation, this study highlights the importance of including surf smelt in ecological and climate change research given the evidence provided here that surf smelt early life stages are affected by stressors associated with near-present climate change and under conditions associated with current nearshore urban development. Given their keystone role in marine ecosystems, reductions to forage fish fitness and population abundance have the potential to cause ecosystem-wide effects. For example, ecosystem models show that Puget Sound seabird populations are negatively affected by reduction in forage fish biomass [5, 54] and piscivorous fish and marine mammals may share similar vulnerability to changes in forage fish abundance [4, 5]. Finally, surf smelt remain one of the

truly understudied forage fish in the north Pacific ecosystem, and further investigation of their potential role in ecosystem response and resilience to global change is warranted.

## Supporting information

**S1 Table. Experimental conditions–embryo heartrate experiment.** Average seawater temperature, pH, $pCO_2$, and $C_T$ for the embryo heartrate experiment. Data are shown as time-averaged means ± 1 SD of (n) measurements. pH and temperature were measured daily, while $C_T$ was measured 3 times per week. $pCO_2$ was derived from temperature, pH, and $C_T$ measurements.
(PDF)

**S1 Fig. Embryo heartrate.** The number of heart beats from embryos (n = 96) per ten seconds from each treatment. Whiskers extend from the upper and lower quartiles to 1.5 times the interquartile range. Data outside of this range are shown as points. Neither temperature nor $pCO_2$ were significant predictors of heartrate.
(TIF)

## Acknowledgments

The authors are grateful for the help of the Washington Department of Fish and Wildlife (T. Sandell). We thank D. Penttila for his guidance on surf smelt collection. We thank the Shannon Point Marine Center for logistical support. We thank B. Miner for statistical and experimental design guidance.

## Author Contributions

**Conceptualization:** Megan Russell, M. Brady Olson, Brooke A. Love.

**Data curation:** Megan Russell, M. Brady Olson, Brooke A. Love.

**Formal analysis:** Megan Russell, M. Brady Olson, Brooke A. Love.

**Funding acquisition:** Megan Russell, M. Brady Olson.

**Investigation:** Megan Russell, M. Brady Olson, Brooke A. Love.

**Methodology:** Megan Russell, M. Brady Olson, Brooke A. Love.

**Project administration:** Megan Russell.

**Resources:** M. Brady Olson, Brooke A. Love.

**Supervision:** M. Brady Olson.

**Visualization:** Megan Russell.

**Writing – original draft:** Megan Russell.

**Writing – review & editing:** M. Brady Olson, Brooke A. Love.

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
