## [Decision Letter · Decision Letter 0]

8 Nov 2021

PONE-D-21-27467Surf smelt accelerate usage of endogenous energy reserves under climate changePLOS ONE

Dear Ms. Russell,

Thank you for submitting your manuscript to PLOS ONE. After careful consideration, we feel that it has merit but does not fully meet PLOS ONE’s publication criteria as it currently stands. Therefore, we invite you to submit a revised version of the manuscript that addresses the points raised during the review process by the reviewers. Note that the comments of Reviewer #1 are also in the attached file.

 In addition to points raised by the reviewers, one confusing aspect of this manuscript is highlighted in the last paragraph of the introduction. It seems clear that the primary hypotheses of this research were related to increased PCO2, however for some reason, increased PCO2 in the introduction changed to total dissolved carbon in the results and discussion sections. Where are the results for the hypotheses that increased PCO2 has effects on these biological factors? It is clearly stated that "The purpose of this study was to investigate the combined effects of elevated seawater temperature and OA on the energy demands of surf smelt embryos and larvae.", yet, none of these results are shown or discussed. The MS (lines 154-160) does a fine job justifying the use of the pH electrode, so why weren't they used for PCO2? At the end of the methods section "CT was the only carbonate parameter used in statistical analyses." is added without any explanation. OA and CT are not synonymous. This confusion in the manuscript needs to be resolved by including the results of the relevant statistical analyses and revising the manuscript accordingly. Sure spectrophotometer pH measurements may be good to the second decimal place and the pH probe might only be good to one decimal place, but you demonstrated the accuracy as noted above. The weak link in your chemical analyses looks to be the refractometer (only one decimal place). I can imagine several ways to approach this (changing/adding figures, results of statistical analyses, clearly stating results of all analyses, etc. in the MS and/or supplemental data). 

This manuscript represents a lot of work, and the results should be very interesting to many people. Please submit your revised manuscript by Dec 23 2021 11:59PM. If you will need more time than this to complete your revisions, please reply to this message or contact the journal office at plosone@plos.org. Please include the following items when submitting your revised manuscript:

We look forward to receiving your revised manuscript.

Kind regards,

Erik V. Thuesen, Ph.D.

Academic Editor

PLOS ONE

Journal Requirements:

2. In your Methods section, please provide additional location information about your collection sites, including geographic coordinates for the data set if available.

The project described in this publication was supported by the Department of the Interior Northwest Climate Adaptation Science Center (NW CASC) through a Cooperative Agreement (G17AC000218) from the United States Geological Survey (USGS). 

The project described in this publication was supported by the Department of the Interior Northwest Climate Adaptation Science Center (NW CASC- https://nwcasc.uw.edu/) through a Cooperative Agreement (G17AC000218) from the United States Geological Survey (USGS). Its contents are solely the responsibility of the authors and do not necessarily represent the views of the NW CASC or the USGS. This manuscript is submitted for publication with the understanding that the United States Government is authorized to reproduce and distribute reprints for Governmental purposes.  The funders had no role in study design, data collection and analysis, decision to publish, or preparation of the manuscript.

Award #:33491

7. Your ethics statement should only appear in the Methods section of your manuscript. If your ethics statement is written in any section besides the Methods, please move it to the Methods section and delete it from any other section. Please ensure that your ethics statement is included in your manuscript, as the ethics statement entered into the online submission form will not be published alongside your manuscript. 

Reviewers' comments:

Reviewer's Responses to Questions

**Comments to the Author**

1. Is the manuscript technically sound, and do the data support the conclusions?

Reviewer #1: Partly

Reviewer #2: Yes

2. Has the statistical analysis been performed appropriately and rigorously? 

Reviewer #1: No

Reviewer #2: I Don't Know

3. Have the authors made all data underlying the findings in their manuscript fully available?

Reviewer #1: Yes

Reviewer #2: Yes

4. Is the manuscript presented in an intelligible fashion and written in standard English?

Reviewer #1: Yes

Reviewer #2: Yes

5. Review Comments to the Author

Reviewer #1: Review for “Surf smelt accelerate usage of endogenous energy reserves under climate change”

This is an interesting manuscript in which the short-term effects of high temperature and high pCO2 exposure on energy reserves of embryos and larvae of surf smelt are described. I have several questions regarding the experimental design and a few remarks for the results section. The discussion is much too long and often discusses topics that are only distantly related to what was actually measured in this study. The authors describe what other studies found but often miss to connect it to their own findings.

Abstract

Line 12-14: One can also argue that because surf smelt are beach spawners, they are exceptionally robust to environmental stress

Line 18:19: Why are you specific with the temperature used but not the pCO2 level? At least give an approximation what “ambient” and what “elevated” means.

Line 26: “CT” has not been explained before use

Line 27: “may impact”. It is more informative if you give a direction. Is it negative or positive impact?

Introduction

Line 107-109: This sentence is repetitive to the previous sentence

Methodology

Line 130: Refer to Table 1.

Line 135-136: Figure 1 is not needed. If you want to show it somewhere put it in the supplementary material

Line 142: Were the embryos shock exposed to the treatment conditions or was there a gradual increase/decrease in water temperature and pCO2?

Line 145-150: Why did you not use the embryos from your embryo experiment to be further studied as larvae? There might be cascading effects from embryo to larva that you missed because of your experimental design which I see as a drawback for your results

Line 172-174: I don’t understand what is meant. Were the temperature treatments not run at the same time?

Line 185-188: What was done to maintain treatment conditions during measurements of heart rates? How long after removal of the embryos from their tanks did it take to start recording the heart rate videos?

Results

Line 206-227: I find this a very weird way of showing the treatment conditions. It gives the impression that you did not exactly know what the treatment conditions were. E.g. line 208: “temperatures remained consistent throughout the duration of the experiment”. If your experiment was planned to use stable temperatures, this is a given. I consider this unnecessary to take up so much space in your manuscript. Referring to Table 1 in your methods section would simply be enough.

Line 211-214: Why is the treatment then called 15C instead of 16C if the water temperature was 16C and not 15C?

Line 213: “summer warming elevated the seawater temperature in the ambient temperature treatment” by how much? For how long?

Line 235-237: Figure 2 does not show this result. I suggest to use a glmer to account for the ratio which cannot be above 1 or below 0. Also I would prefer showing the error as 95% intervals and offsetting the CT data for clarity of the error bars. Maybe this would then visually support the result that pCO2 had an effect. Also you describe that data from the 13C treatment are missing in line 173 (which I commented on before) but why is there a data point for day 13 for the ambient pCO2 treatment but not for the elevated pCO2 treatment?

Line 254: In Figure 4 it looks like larvae used for the 18C treatment started off with smaller yolk areas already. How can this be and was the difference in yolk area size at the start accounted for when comparing the yolk area at the final day?

Discussion

Line 277-285: Can you draw some conclusion from all these Q10 values? The Q10 that you calculated is smaller than the average 2.9 calculated by Heming and Buddington. If you are using this reference, it indicates that surf smelt are less temperature dependent than most fish species studied (in case this is the point which you want to make by referring to Heming and Buddington). Why are you stating this reference? What is the point that you want to make with it? This is especially true for the last sentence in this paragraph (line 283-285).

Line 286-292: Similar to the previous paragraph. Which point do you want to make with the review of Villalobos et al 2020? We do not know the temperature performance curve or at least the temperatures at which C. pallasi occurs in nature.

Line 310 to 322: Although there is nothing wrong with this paragraph I question why it is included. You mention that you did not measure developmental rate and you don’t know if a faster development due to higher temperature is present in surf smelt, so why do you then include this paragraph in your manuscript?

Line 337: Which temperatures are considered “temperature stress” in surf smelt larvae?

Line 338-342: What do these sentences add to your story?

Line 354-363: Again, why is this paragraph included? This paragraph is about acid-base regulation which was not part of what you have measured. You are discussing topics that you did not measure in your study.

Line 370-374: Did your embryos and larvae come from adults that were exposed to high pCO2? Relate the studies that you refer to to your own study.

Line 383-388: Also have a look at Leo et al 2018. They used herring from Norway and found higher susceptibility of embryos and larvae to high pCO2 than Sswat and Franke. But also here, the point that you want to make by referring to these herring studies is not really clear.

Line 388-390: I don’t think that you can draw this conclusion by studying surf smelt for only 3 days.

Line 415-424: While this all is correct I think it is too far off from what you did in your study and does not need to be included here.

Line 425-465: These three paragraphs are great and if they are filled out with your main points from your results they would make a great discussion making most of the previous discussion (line 264-424) redundant

Reviewer #2: General comments:

The manuscript (PONE-D-21-27467) titled: “Surf smelt accelerate usage of endogenous energy reserves under climate change” aimed to determine if energy reserves of surf smelt embryos and larvae are affected by single and interactive effect of ocean warming and ocean acidification. To address this aim, before reaching 24 hours old, surf smelt were acclimated to a combination of three temperatures 13, 15, and 18°C and two treatment of total carbon (CT). Yolk sac and globule size, embryo and larval size, and heart rates were then compared between the 6 treatment groups (3 temperature x 2 CT). The main conclusion of the study is that temperature but not CT affect energy reserves in surf smelt early stages and that the energy demand increase with the combination of the two stressors.

Overall the manuscript is clear and well written. Over the past decade a number of studies addressed the impact of future warming and pCO2 levels on fish metabolism but only recently, studies have begun to address how ocean acidification and warming affect the metabolism of fish embryos. While the importance of studying surf smelt is well justify, reasons for investigating warming and CT are unclear is this manuscript and make partly sense only once reading the “Ecological Implications” section at the end of the discussion. Similarly, the choice for the experimental conditions lacks in justification and supporting statement. The following are additional suggestions for improvement.

Major changes:

Lines 59-77: The discussion state (lines 427-430) that human activities have induce a reduction in vegetation in surf smelt spawning grounds leading to an increase in temperature due to the lack of shading. A few sentences in the introduction would be useful to justify the importance of investigating the effect of temperature stress on surf smelt.

Lines 78-84: Similarly here, a justification for studying the effect of ocean acidification is missing. It appears only in the discussion lines 444-446. I think it would be benefic to reshape the introduction for better understand the basis of the study.

Lines 95-97: Is it possible to provide the range of pCO2 encounter in the northeast Pacific Ocean systems? What are the values predicted by Feely et al. (2020) for the Puget Sound ecosystem and how do they compare with predictions from climate model forced with IPCC?

Lines 125-127: How much the temperature fluctuates along the Salish Sea shoreline? It would be good to add this information to provide context on the choice of acclimation temperatures (13, 15, 18°C). The justification for using 15 and 18°C is lacking. Presumably these represent future climate conditions encounter in shallow Salish Sea waters during the summer but according to which climate model and how far in the future?

Line 204: Is there a possibility to mention/display mortality and hatching success? Were there any difference between treatments?

Lines 211-214: What was the temperature of the water at time of collection? For how long was the temperature changed from 15°C to 16°C? Why choose 16°C, did the ambient condition only increase of 1°C (13 to 14°C) during the summer? Then would using a delta be beneficial: Δ2°C from the “ambient” condition. Thus, if temperature of the ambient condition increased to 14°C during the summer the medium condition was always 2°C above. Was the 18°C condition always maintain at 18°C?

Lines 223-224: Only the average standard deviation is displayed, it would be a nice reminder of the condition and for better flow to have the value presented as: mean ±SD. In addition, it is highlighted here that variability in pCO2 was higher for the high pCO2 condition. It would be interesting to mention it in the discussion. How higher variability (compare to more stable treatment) impacted the results? What is usually observed in the wild?

Lines 299-300: Tradeoffs is a hypothesis that may explained the results but results can also be explained by the multiple performances –multiple optima (MPMO) (Clark et al. 2013), stating that different physiological activities have potentially different thermal optima. While this theory have been developed in regards to performance traits of developing larvae it may be applicable in embryos.

Lines 355-356: In accordance with the discussion here, a recent study (Dahlke et al. 2020), demonstrated that early-life stages without functional gills appear to be better equipped in terms of ion homeostasis than previously thought.

Lines 375-381: It would be interesting to link further the behavior/biological cycle of surf smelt with their capacity to withstand elevated CT. The author might want to look into the “Ocean variability hypothesis” which connect the duration and distance of the stay inshore of migratory fish with their resilience to CO2 (Baumann, 2019).

Lines 426-447: The effort to discuss the ecological implications of the results is to be applauded, however, I believe that an important part of this section should be moved to the introduction to justify why this study focus on the two stressors: temperature and pCO2.

Minor changes

Line 39: What is implied by “clean, cold water and unaltered shoreline”? It would be beneficial to provide thermal range for surf smelt and maybe and index of water quality.

Line 58: A reference is missing to support this statement.

Line 173: It is not clear why ambient temperature treatment only lasted for 10 days.

Line 183: A reference is missing to support this statement.

Results: Were there any basins effects? While the details of statistic test are provided in the table, it would be engaging to have the p-value in the text.

Lines 229-237: How the combination of the stressors affected yolk usage?

Line 354-355: “despite the very high concentration”, again we have no indication of the concentration find in the wild. What is it normally observed in their habitat? Why is it considered “very high”?

Figures: Indication of significant differences using symbols (*) or letters would be benefic.

References:

Clark TD, Sandblom E, Jutfelt F. Aerobic scope measurements of fishes in an era of climate change: respirometry, relevance and recommendations. J Exp Biol. 2013;216(15):2771–2782.

Dahlke FT, Leo E, Mark FC, Pörtner HO, Bickmeyer U, Frickenhaus S, Storch D. Effects of ocean acidification increase embryonic sensitivity to thermal extremes in Atlantic cod, Gadus morhua. Glob Chang Biol. 2017 Apr;23(4):1499-1510.

Baumann, H. (2019). Experimental assessments of marine species sensitivities to ocean acidification and co-stressors: How far have we come? 1. Canadian Journal of Zoology, 97.

6. PLOS authors have the option to publish the peer review history of their article (what does this mean?). If published, this will include your full peer review and any attached files.

Reviewer #1: No

Reviewer #2: No

---

## [Author Response · Author response to Decision Letter 0]

20 Jan 2022

Editor Comments:

In addition to points raised by the reviewers, one confusing aspect of this manuscript is highlighted in the last paragraph of the introduction. It seems clear that the primary hypotheses of this research were related to increased PCO2, however for some reason, increased PCO2 in the introduction changed to total dissolved carbon in the results and discussion sections. Where are the results for the hypotheses that increased PCO2 has effects on these biological factors? It is clearly stated that "The purpose of this study was to investigate the combined effects of elevated seawater temperature and OA on the energy demands of surf smelt embryos and larvae.", yet, none of these results are shown or discussed. The MS (lines 154-160) does a fine job justifying the use of the pH electrode, so why weren't they used for PCO2? At the end of the methods section "CT was the only carbonate parameter used in statistical analyses." is added without any explanation. OA and CT are not synonymous. This confusion in the manuscript needs to be resolved by including the results of the relevant statistical analyses and revising the manuscript accordingly. Sure spectrophotometer pH measurements may be good to the second decimal place and the pH probe might only be good to one decimal place, but you demonstrated the accuracy as noted above. The weak link in your chemical analyses looks to be the refractometer (only one decimal place). I can imagine several ways to approach this (changing/adding figures, results of statistical analyses, clearly stating results of all analyses, etc. in the MS and/or supplemental data). 

We appreciate your feedback on the clarity of our approach to which carbonate system parameters are used in the analysis and reporting of the study. We recognize that pCO2 is the primary parameter used in most OA discussions, and there are good physical and experimental reasons for this. Many systems (including the atmosphere/ocean system writ large) do rely on gas exchange, and so pCO2 is the primary parameter to encapsulate OA. However, the design of our system is such that total carbon is the carbonate parameter we are most directly manipulating as CO2 gas is quantitatively dissolved on introduction to the mixing tanks. Therefore, we were careful to state our broad hypothesis in terms of OA and our more specific ones in terms of CT: 

The purpose of this study was to investigate the combined effects of elevated seawater temperature and OA on the energy demands of surf smelt embryos and larvae…We hypothesized that energy demand, measured as embryo heart rate and yolk sac/oil globule exhaustion, for both embryos and larvae would increase under elevated seawater temperature and CT (i.e. acidification). It was also predicted that the highest energy usage would be observed under simultaneous increased temperature and elevated CT due to the additive effect of these climate stressors on compensatory homeostatic processes.

We also chose total carbon as the primary carbonate variable in our statistical analysis because pCO2 is highly temperature dependent. The use of total carbon rather than pCO2 maintains the least amount of correlation between potential variables which would make it more difficult to disentangle the effects of carbonate system changes versus temperature changes. 

Clearly, we can do better in making a case for total carbon as the most appropriate measure of OA in this instance. In order to address your concerns in this area, added some text in the methods section with recognition that pCO2 is the parameter most commonly used to represent the suite of changes associated with ocean acidification and making our choice to use total carbon in the analysis here more clear. This explanation emphasizes the design of the system and the co-variation of pCO2 and temperature, which is not desirable when we want to understand the effects of both temperature and OA related changes. We have reviewed the introduction and discussion to be sure we are careful about where the use of OA, pCO2 and total carbon are appropriate. We could also bring more attention to the pCO2 values that we did calculate in the results and discussion if you recommend that we do so. 

Additional revisions we could make include adding data about the correlations that exist between pCO2, total carbon and temperature in our data, to make the point that total carbon does a good job of representing OA, and is highly correlated with pCO2, but that it is less correlated with temperature, which is desirable in this situation for the statistical analysis. This might be more appropriate in supplementary information if it is needed. While salinity does not have a strong effect on pCO2 with the kind of variability observed in this study, there are high quality monitors on the seawater system and we could retrieve those data if necessary. However, the variability in the pH numbers will swamp any minor salinity effects, so we suggest this step may not be necessary. 

Journal Requirements:

These documents were reviewed and the format was adjusted accordingly. 

2. In your Methods section, please provide additional location information about your collection sites, including geographic coordinates for the data set if available.

We added the specific name of the Bay that samples were collected as well as coordinates, and this can be referenced with any available regional map.

This project was conducted in collaboration with the Washington State Department of Fish and Wildlife (WDFW) forage fish team. Embryos were collected under their auspices and, as such, we did not need our own WDFW collection permit. 

This was reviewed and ensured to match.

The project described in this publication was supported by the Department of the Interior Northwest Climate Adaptation Science Center (NW CASC) through a Cooperative Agreement (G17AC000218) from the United States Geological Survey (USGS). 

We moved this section to the ‘Funding’ section. We hope this rectifies the issue.

The project described in this publication was supported by the Department of the Interior Northwest Climate Adaptation Science Center (NW CASC- https://nwcasc.uw.edu/) through a Cooperative Agreement (G17AC000218) from the United States Geological Survey (USGS). Its contents are solely the responsibility of the authors and do not necessarily represent the views of the NW CASC or the USGS. This manuscript is submitted for publication with the understanding that the United States Government is authorized to reproduce and distribute reprints for Governmental purposes. The funders had no role in study design, data collection and analysis, decision to publish, or preparation of the manuscript.

Award #:33491

The data has been published in the SENOE repository, citation:

Russell Megan (2018). Surf Smelt Embryo and Larvae Data. SEANOE. https://doi.org/10.17882/85830

Our cover letter has been updated to reflect this change.

7. Your ethics statement should only appear in the Methods section of your manuscript. If your ethics statement is written in any section besides the Methods, please move it to the Methods section and delete it from any other section. Please ensure that your ethics statement is included in your manuscript, as the ethics statement entered into the online submission form will not be published alongside your manuscript. 

This section has been moved to the Methods section of the manuscript

Reviewer #1 Comments/responses

Reviewer #1: Review for “Surf smelt accelerate usage of endogenous energy reserves under climate change”

This is an interesting manuscript in which the short-term effects of high temperature and high pCO2 exposure on energy reserves of embryos and larvae of surf smelt are described. I have several questions regarding the experimental design and a few remarks for the results section. The discussion is much too long and often discusses topics that are only distantly related to what was actually measured in this study. The authors describe what other studies found but often miss to connect it to their own findings.

We appreciate this comment. This manuscript describes the M.S. thesis work of the lead author. We recognize that the rather long discussion in the original manuscript reflects this style. In the revision we cut the discussion down from seven pages to just over three pages, and have removed all components that we believe only peripherally supported our findings. We believe this change in response to your suggestion makes for a much stronger discussion.

Abstract

Line 12-14: One can also argue that because surf smelt are beach spawners, they are exceptionally robust to environmental stress

Excellent point. We changed this sentence to reflect that as obligate beach spawners, they are exposed to both marine and terrestrial stressors. We also highlighted this argument in the discussion.

Line 18:19: Why are you specific with the temperature used but not the pCO2 level? At least give an approximation what “ambient” and what “elevated” means.

We amended this to show specificity with total carbon and pCO2.

Line 26: “CT” has not been explained before use

Thank you for letting us know. It is now defined on line 19. 

Line 27: “may impact”. It is more informative if you give a direction. Is it negative or positive impact?

We amended this sentence to reflect simply the direction of change. 

Introduction

Line 107-109: This sentence is repetitive to the previous sentence

We meant for the first sentence to reference these stressors as isolated, and in the following sentence as occurring simultaneously. We amended the sentence to make this more clear. 

Methodology

Line 130: Refer to Table 1.

Thank you for this suggestion. We added this reference.

Line 135-136: Figure 1 is not needed. If you want to show it somewhere put it in the supplementary material

Excellent suggestion. We removed this figure from the manuscript.

Line 142: Were the embryos shock exposed to the treatment conditions or was there a gradual increase/decrease in water temperature and pCO2?

The embryos were not slowly acclimated to these conditions. 

Line 145-150: Why did you not use the embryos from your embryo experiment to be further studied as larvae? There might be cascading effects from embryo to larva that you missed because of your experimental design which I see as a drawback for your results

We agree that this is a shortcoming, but did not have the manpower to include this design.

Line 172-174: I don’t understand what is meant. Were the temperature treatments not run at the same time?

They were run at the same time, but the ambient treatment was terminated after 10 days because the natural warming of the seawater brought the temperature close to the medium temperature treatment. As such, we lost temperature separation between treatments at 10 days. We amended this section for clarity. 

Line 185-188: What was done to maintain treatment conditions during measurements of heart rates? How long after removal of the embryos from their tanks did it take to start recording the heart rate videos?

The embryos were moved individually in their treatment water and placed immediately under the scope. In preliminary experiments we did test to make sure that temperature did not increase or decrease during the short 10-30 seconds of filming. 

Results

Line 206-227: I find this a very weird way of showing the treatment conditions. It gives the impression that you did not exactly know what the treatment conditions were. E.g. line 208: “temperatures remained consistent throughout the duration of the experiment”. If your experiment was planned to use stable temperatures, this is a given. I consider this unnecessary to take up so much space in your manuscript. Referring to Table 1 in your methods section would simply be enough.

We agree, and have cut much of this section.

Line 211-214: Why is the treatment then called 15C instead of 16C if the water temperature was 16C and not 15C?

This sentence is a mistake, and thank you for catching it. They were adjusted from 14C to 15C. This section is part of the passage that is now omitted.

Line 213: “summer warming elevated the seawater temperature in the ambient temperature treatment” by how much? For how long?

This fact was addressed in your earlier comment about lines 172-174, and was the reason that the ambient temperature treatment was stopped at 10 days.

Line 235-237: Figure 2 does not show this result. I suggest to use a glmer to account for the ratio which cannot be above 1 or below 0. Also I would prefer showing the error as 95% intervals and offsetting the CT data for clarity of the error bars. Maybe this would then visually support the result that pCO2 had an effect. Also you describe that data from the 13C treatment are missing in line 173 (which I commented on before) but why is there a data point for day 13 for the ambient pCO2 treatment but not for the elevated pCO2 treatment?

We analyzed the data now with a generalized linear model (family= binomial, link=logistic) to account for the ratio and give the model stronger power. Results have been modified to reflect this model change. We modified the figure to offset the data points and displayed error as standard error to allow for greater clarity in the figure. The data were binned based on temperature. There was an experimental basin in the ambient pCO2 treatment that remained in the 13C temperature bin, but there was not one for the elevated PCO2 treatment. 

Line 254: In Figure 4 it looks like larvae used for the 18C treatment started off with smaller yolk areas already. How can this be and was the difference in yolk area size at the start accounted for when comparing the yolk area at the final day?

The data start in Figure 4 on day one. That is, larvae were in treatment water for 24 hours prior to the first measurement. Thus, the 18C treatments had already shown evidence of accelerated yolk usage. Unfortunately, we do not have time zero measurements. The reduction in yolk size expressed in lines 252-253 does take into account the original size in each treatment. The average size at the end does not (lines 253-254) and, because of this, has been removed. We are grateful you caught that error.

Discussion

Line 277-285: Can you draw some conclusion from all these Q10 values? The Q10 that you calculated is smaller than the average 2.9 calculated by Heming and Buddington. If you are using this reference, it indicates that surf smelt are less temperature dependent than most fish species studied (in case this is the point which you want to make by referring to Heming and Buddington). Why are you stating this reference? What is the point that you want to make with it? This is especially true for the last sentence in this paragraph (line 283-285).

We appreciate this comment, and now view it as a missed opportunity to elaborate on an earlier point that you made (…One can also argue that because surf smelt are beach spawners, they are exceptionally robust to environmental stress). We amended this section to include a short discussion of the relevance of our lower observed Q10 values.

Line 286-292: Similar to the previous paragraph. Which point do you want to make with the review of Villalobos et al 2020? We do not know the temperature performance curve or at least the temperatures at which C. pallasi occurs in nature.

Upon reflection, we see your point and agree that this adds little to no value to the discussion of our results. We have omitted this section.

Line 310 to 322: Although there is nothing wrong with this paragraph I question why it is included. You mention that you did not measure developmental rate and you don’t know if a faster development due to higher temperature is present in surf smelt, so why do you then include this paragraph in your manuscript?

This section has been removed from the manuscript.

Line 337: Which temperatures are considered “temperature stress” in surf smelt larvae?

Your point is well taken, and we amended this sentence.

Line 338-342: What do these sentences add to your story?

We amended this section, but believe accelerated use of the oil globule will, by way of its influence on fish buoyancy, negatively affect the ability of larval fish to capture prey. Thus, we are connecting our results to some ecological implications.

Line 354-363: Again, why is this paragraph included? This paragraph is about acid-base regulation which was not part of what you have measured. You are discussing topics that you did not measure in your study.

This section has been removed from the manuscript.

Line 370-374: Did your embryos and larvae come from adults that were exposed to high pCO2? Relate the studies that you refer to your own study.

This section has been removed from the manuscript.

Line 383-388: Also have a look at Leo et al 2018. They used herring from Norway and found higher susceptibility of embryos and larvae to high pCO2 than Sswat and Franke. But also here, the point that you want to make by referring to these herring studies is not really clear.

This section has been removed from the manuscript.

Line 388-390: I don’t think that you can draw this conclusion by studying surf smelt for only 3 days.

This section has been removed from the manuscript.

Line 415-424: While this all is correct I think it is too far off from what you did in your study and does not need to be included here.

This section has been removed from the manuscript.

Line 425-465: These three paragraphs are great and if they are filled out with your main points from your results they would make a great discussion making most of the previous discussion (line 264-424) redundant

As previously stated, we restructured our discussion based on this comment. In doing so, it is considerably shorter and we believe more directly aligns with our findings.

Reviewer #2 Comments/responses

The manuscript (PONE-D-21-27467) titled: “Surf smelt accelerate usage of endogenous energy reserves under climate change” aimed to determine if energy reserves of surf smelt embryos and larvae are affected by single and interactive effect of ocean warming and ocean acidification. To address this aim, before reaching 24 hours old, surf smelt were acclimated to a combination of three temperatures 13, 15, and 18°C and two treatment of total carbon (CT). Yolk sac and globule size, embryo and larval size, and heart rates were then compared between the 6 treatment groups (3 temperature x 2 CT). The main conclusion of the study is that temperature but not CT affect energy reserves in surf smelt early stages and that the energy demand increase with the combination of the two stressors.

Overall the manuscript is clear and well written. Over the past decade a number of studies addressed the impact of future warming and pCO2 levels on fish metabolism but only recently, studies have begun to address how ocean acidification and warming affect the metabolism of fish embryos. 

While the importance of studying surf smelt is well justify, reasons for investigating warming and CT are unclear is this manuscript and make partly sense only once reading the “Ecological Implications” section at the end of the discussion. Similarly, the choice for the experimental conditions lacks in justification and supporting statement. The following are additional suggestions for improvement.

We have made considerable effort to address this concern throughout the manuscript. 

Major changes:

Lines 59-77: The discussion state (lines 427-430) that human activities have induce a reduction in vegetation in surf smelt spawning grounds leading to an increase in temperature due to the lack of shading. A few sentences in the introduction would be useful to justify the importance of investigating the effect of temperature stress on surf smelt.

We appreciate your comment and have added a section highlighting this in the introduction.

Lines 78-84: Similarly here, a justification for studying the effect of ocean acidification is missing. It appears only in the discussion lines 444-446. I think it would be benefic to reshape the introduction for better understand the basis of the study.

We have added a section in the introduction justifying the OA component of this study. 

Lines 95-97: Is it possible to provide the range of pCO2 encounter in the northeast Pacific Ocean systems? What are the values predicted by Feely et al. (2020) for the Puget Sound ecosystem and how do they compare with predictions from climate model forced with IPCC?

We have added a section to the introduction addressing this concern.

Lines 125-127: How much the temperature fluctuates along the Salish Sea shoreline? It would be good to add this information to provide context on the choice of acclimation temperatures (13, 15, 18°C). The justification for using 15 and 18°C is lacking. Presumably these represent future climate conditions encounter in shallow Salish Sea waters during the summer but according to which climate model and how far in the future?

We have added considerable justification for using these temperatures. 

Line 204: Is there a possibility to mention/display mortality and hatching success? Were there any difference between treatments?

We did not measure this as we were focusing on energy utilization and were limited by person hours.

Lines 211-214: What was the temperature of the water at time of collection? 

13℃

For how long was the temperature changed from 15°C to 16°C? Why choose 16°C, did the ambient condition only increase of 1°C (13 to 14°C) during the summer? Then would using a delta be beneficial: Δ2°C from the “ambient” condition. Thus, if temperature of the ambient condition increased to 14°C during the summer the medium condition was always 2°C above. Was the 18°C condition always maintain at 18°C?

This passage, based on comments from another reviewer, has been cut. The original passage had an error. Namely, the moderate treatment was not moved up to 16℃, but from 14℃ to 15℃. We are referring all readers to table 1.

Lines 223-224: Only the average standard deviation is displayed, it would be a nice reminder of the condition and for better flow to have the value presented as: mean ±SD. In addition, it is highlighted here that variability in pCO2 was higher for the high pCO2 condition. It would be interesting to mention it in the discussion. How higher variability (compare to more stable treatment) impacted the results? What is usually observed in the wild?

We agree that presenting standard deviations without the means is a little awkward and makes them less meaningful. Because pCO2 is temperature sensitive, we do not want to repeat a large number of mean values (18 in total) already presented in the data table and referenced. 

Regarding the effect of variability, this pattern is very common in OA studies and is not usually included in the analysis of results. These animals are exposed to high variability in their natural spawning habitat, so large physiological effects associated with these difference in variability seem unlikely. 

Lines 299-300: Tradeoffs is a hypothesis that may explained the results but results can also be explained by the multiple performances –multiple optima (MPMO) (Clark et al. 2013), stating that different physiological activities have potentially different thermal optima. While this theory have been developed in regards to performance traits of developing larvae it may be applicable in embryos.

Another reviewer was very critical of the length of our original discussion, and the weak connection to our own data. As such, and as you will see, we drastically reduced the length of our original discussion. In doing so, this section was cut from the manuscript.

Lines 355-356: In accordance with the discussion here, a recent study (Dahlke et al. 2020), demonstrated that early-life stages without functional gills appear to be better equipped in terms of ion homeostasis than previously thought.

This section has been removed from the manuscript.

Lines 375-381: It would be interesting to link further the behavior/biological cycle of surf smelt with their capacity to withstand elevated CT. The author might want to look into the “Ocean variability hypothesis” which connect the duration and distance of the stay inshore of migratory fish with their resilience to CO2 (Baumann, 2019).

This is an excellent suggestion. We added to this discussion, including pH variability at the site of collection and how this relates to the OVH Hypothesis.

Lines 426-447: The effort to discuss the ecological implications of the results is to be applauded, however, I believe that an important part of this section should be moved to the introduction to justify why this study focus on the two stressors: temperature and pCO2.

We added justification to the introduction and re-framed our discussion around these ecological implications. 

Minor changes

Line 39: What is implied by “clean, cold water and unaltered shoreline”? It would be beneficial to provide thermal range for surf smelt and maybe and index of water quality.

We do not have or can find a thermal range for surf smelt, nor an index of water quality required for spawning. We removed the text associated with ‘clean’ and ‘cold’ seawater. We did, however, leave the in reference to ‘unaltered’ as this relates to our discussion of modified beaches, and the references we cite support that statement.

Line 58: A reference is missing to support this statement.

Thank you for alerting us to this omission. We have added a recent reference to support this statement.

Line 173: It is not clear why ambient temperature treatment only lasted for 10 days.

We amended this section to add clarity.

Line 183: A reference is missing to support this statement.

A reference was added for this statement (Thorarensen et al. 1996).

Results: Were there any basins effects? While the details of statistic test are provided in the table, it would be engaging to have the p-value in the text.

Basins were included as a random variable and none were observed. We understand your point about adding p values in the text, but we made the decision to keep them out and refer the reader to the tables. This way the text is more readable.

Lines 229-237: How the combination of the stressors affected yolk usage?

Stressor interactions were investigated when determining the best fit model. All interactions not shown were not significant and including them lowered the overall fit of the model based on AIC score comparisons. 

Line 354-355: “despite the very high concentration”, again we have no indication of the concentration find in the wild. What is it normally observed in their habitat? Why is it considered “very high”?

As stated previously, we added significant text in the introduction informing the reader as to OA concentrations surf smelt experience in the study region.

Figures: Indication of significant differences using symbols (*) or letters would be beneficial.

Our figures are not a display of our models, but of the raw data that was used to create these models. Because we looked at overall trends over time rather than looking at individual time points, it would not make sense to annotate significant differences on our figures at specific time points.

All of our figures were uploaded to PACE and checked.

---

## [Decision Letter · Decision Letter 1]

10 Feb 2022

PONE-D-21-27467R1Surf smelt accelerate usage of endogenous energy reserves under climate changePLOS ONE

Dear Megan Russell,

Thank you for re-submitting your manuscript to PLOS ONE. After careful consideration, we feel that it has merit but does not fully meet PLOS ONE’s publication criteria as it currently stands. Therefore, we invite you to submit a revised version of the manuscript that addresses the points raised during the review process. Thank you for addressing concerns of the two first reviewers. Your response to my previous comments about confusing total carbon and pCO2 was not adequate, and I solicited another reviewer's opinion. You will need to re-write the manuscript with regards to pCO2 instead of total carbon in order for it to be acceptable (cf. comment #6 of Reviewer #3). Please look over the comments of reviewer #3 carefully. You should find them useful in your data analyses and discussion. Considering those points should further improve your manuscript.

We look forward to receiving your revised manuscript.

Kind regards,

Erik V. Thuesen, Ph.D.

Academic Editor

PLOS ONE

Reviewers' comments:

Reviewer's Responses to Questions

**Comments to the Author**

1. If the authors have adequately addressed your comments raised in a previous round of review and you feel that this manuscript is now acceptable for publication, you may indicate that here to bypass the “Comments to the Author” section, enter your conflict of interest statement in the “Confidential to Editor” section, and submit your "Accept" recommendation.

Reviewer #3: (No Response)

2. Is the manuscript technically sound, and do the data support the conclusions?

Reviewer #3: Partly

3. Has the statistical analysis been performed appropriately and rigorously? 

Reviewer #3: I Don't Know

4. Have the authors made all data underlying the findings in their manuscript fully available?

Reviewer #3: Yes

5. Is the manuscript presented in an intelligible fashion and written in standard English?

Reviewer #3: Yes

6. Review Comments to the Author

Reviewer #3: This paper poses that beach spawning forage fish, smelt, may be affected by increasing temperature and seawater CO2 and possible interactions between these variables. The question is interesting, but tricky. I’ve provided some additional things for the authors to consider below. The accelerated use of yolk at high temperatures is clear but not unexpected. The effects of CO2 and interactions with temperature are minimal/none. The paper is generally well-written.

See attachment for further comments and figures.

7. PLOS authors have the option to publish the peer review history of their article (what does this mean?). If published, this will include your full peer review and any attached files.

Reviewer #3: No

---

## [Author Response · Author response to Decision Letter 1]

25 Mar 2022

Response to Reviewers

Editors Comments

Thank you for addressing concerns of the two first reviewers. Your response to my previous comments about confusing total carbon and pCO2 was not adequate, and I solicited another reviewer's opinion. You will need to re-write the manuscript with regards to pCO2 instead of total carbon in order for it to be acceptable (cf. comment #6 of Reviewer #3). Please look over the comments of reviewer #3 carefully. You should find them useful in your data analyses and discussion. Considering those points should further improve your manuscript.

We have rewritten the manuscript focusing on pCO2 rather than total carbon treatments, including re-running the statistics using pCO2 as a continuous variable. We hope this change and modification to the subsequent manuscript fulfills this request.

Reviewer Comments

1. The eggs are found buried in sediment on the beach and its unclear how much time is spent immersed in seawater. At lower tides, the eggs are in a moist environment that may heat up due to sun exposure. I imagine the CO2 levels in the sediment increase at this point relative to that found in seawater. No measurements of pH or CO2 in the sand were made. If CO2 is higher in seawater than in the emmersed sand, then one would expect high CO2 to cooccur with lower (ambient) temperatures. However, the only interaction between CO2 and temperature found was at high temperature and high CO2 (but see point 2). 

We understand that our experimental design does not replicate the dynamic nature of embryo exposure, and we acknowledge this in lines 147-151. 

2. Heart rate was highly variable but not significantly different across temperatures. The high Temp/high CO2 treatment was significantly lower but the error bars are enormous and completely overlap with the other treatments. So that result is not compelling. The text states that heart rate increased at high temp/ambient CO2, but that is not significant and so should be changed (no effect). 

When re-running the stats with pCO2 as a continuous variable, there was no significant effect of temperature or pCO2 on heartrate. The results and discussion have been modified to reflect this. The table showing experimental conditions for the heartrate experiment and the figure showing the data have been moved to supplemental materials. 

3. The text makes the case that the reduced heart rate may have resulted from CO2/pH effects on oxygen binding in the blood. However, this would only be important if the embryos were near maximum metabolic rate for the available oxygen (i.e. near the Pcrit for whatever their oxygen consumption rate was). Aerobic scope (difference between max and rest metabolic rates) is largely unknown for eggs. However, Pcrits for fish eggs below 50% saturation have been reported. In that case, any reduction in oxygen transport would not translate into a reduced metabolic rate under the oxic experimental conditions. Something to ponder.

This possible mechanism was suggested in a source, see below. However, your points are valid and due to the unknown nature of this mechanism and wanting to avoid speculation, we have removed this sentence. 

Esbaugh A (2018) Physiological implications of ocean acidification for marine fish: emerging patterns and new insights. J Comp Physiol B 188:1-13. doi:10.1007/s00360-017-1105-6

 4. Heart rate would be expected to increase with metabolic rate, which nearly always increases with temperature. An increased metabolic rate is consistent with greater consumption of yolk and fat stores at high temperature. Its unclear why heart rate wouldn’t increase at high temperature except that its so variable as to be undetectable. 

We agree and have largely removed all discussion of heart rate due, as stated above, to the fact that when we re-analyzed our data using pCO2 as a continuous variable, we did not observe a significant effect of temp or pCO2 on heart rate. And as you say, with variable data and our small sample size, it is unlikely to yield observable significant result.

5. There is a small, but likely significant, “experiment” effect on the CO2 levels achieved (see figure below). The embryo heart rate experiment had the highest PCO2 levels achieved of any experiment. Its not possible to say whether than influenced the outcome relative to the slightly lower values in the other two experiments. 

We understand this could be an effect but as you said, it is not possible to say whether this influenced the outcome. 

6. As pointed out by the editor, the paper discusses PCO2 in the introduction and frames hypotheses about the effects of PCO2 on biology. However, the authors instead perform statistical analyses on total carbon. As stated on line 124: “While pCO2 is the parameter most commonly used to represent the suite of changes associated with OA, and is the parameter actually manipulated in many systems, CT is the control variable in this system.” It is further stated that PCO2 is temperature sensitive and this is why CT was used instead. However, both variables are temperature sensitive. In fact, what changes with temperature is the solubility of CO2. If a given amount of CO2 is dissolved in seawater, both total carbon and carbon dioxide pressure will change and this change will be different at different temperatures because of changing solubility with temperature. Cold water will hold more of the CO2 (thus higher CT) and the dissolved CO2 will exert less pressure. This is seen clearly in a graphical representation of the data (below), especially within the “ambient” treatment. So ultimately it doesn’t matter which is stated throughout the text…the treatment has high total CO2 and high PCO2 relative to the ambient control. However, because the range of values is different for each variable, the statistical analyses could be influenced. For example, if the PCO2 doubles, the pH will decrease by 0.3 units and total CO2 increases by only a small fraction because of the large amount of CO2 already dissolved in seawater. Thus analyzing the interacting effects of CT and temperature may provide less clear results than analyzing the interacting effects of PCO2 and temperature…even though they are two sides of the same coin. Because CO2 was added at a particular rate, rather than a particular amount or to achieve a particular PCO2, pH or TC, the various carbonate parameters all change with temperature in interrelated ways and the different experiments have different values.

We agree with your summation and have re-analyzed the data using pCO2 as a continuous variable to look at pCO2 interactions. The only change to our results was the loss of statistical significance in the heartrate experiment. As a result, the manuscript has been modified to reflect these changes.

---

## [Editor Report · Decision Letter 2]

18 Apr 2022

PONE-D-21-27467R2Surf smelt accelerate usage of endogenous energy reserves under climate changePLOS ONE

Dear Dr. Russell,

Thank you for re-submitting your manuscript to PLOS ONE. After careful consideration, we feel that it has merit but does not fully meet PLOS ONE’s publication criteria as it currently stands. Therefore, we invite you to submit a revised version of the manuscript that addresses the points below.

1) The manuscript is still confusing pCO2 and CT. For example, in the abstract the sentence "two total carbon (CT) treatments" makes it seem like the treatments were made in a fashion other than manipulating pCO2. In fact, pCO2 was used to manipulate the system, and that should be make clear throughout the manuscript. Everyone knows that when you change pCO2, CT also changes. That sentence in the abstract should be re-written to accurately reflect the experiments. Other places that need attention 2) Line 117: 20-lb3) lines 133 and 134: see comment number 1 above.4) Lines 146 and 156: 200-mL5) Lines 147-151: This sentence is awkward. Maybe break it up into 2 or three sentences?6) Line 152: This sentence is awkward. Maybe explicitly state this is a different experiment to clarify this for the reader.7) Line 153: 12-L8) Line 167: 20-mL9) Line 182: I suggest changing to ... 'into a 6-well plate labeled with its corresponding treatment.'10) Please give citation for ImageJ11) Lines 189-194. This paragraph is a bit difficult to follow. I suggest re-writing it.12) Lines 202-205 can be deleted as they appear on lines 220-222. Add the WA DFW info to the approval statement, too.13) In the statistical analysis section of the methods, please state which stats software was used.14) Lines 225-239 and anywhere else: Please see comment number one above.15) Table 3: Please correct the p for pCO2. It now reads 0.0.541816) A recent paper by Murray and Klinger (J Exp Biol (2022) 225 (5): jeb243501) is applicable to your study and could be a good addition to your discussion.

We look forward to receiving your revised manuscript.

Kind regards,

Erik V. Thuesen, Ph.D.

Academic Editor

PLOS ONE
---

## [Author Response · Author response to Decision Letter 2]

1 Jun 2022

Response to Reviewers

1) The manuscript is still confusing pCO2 and CT. For example, in the abstract the sentence "two total carbon (CT) treatments" makes it seem like the treatments were made in a fashion other than manipulating pCO2. In fact, pCO2 was used to manipulate the system, and that should be make clear throughout the manuscript. Everyone knows that when you change pCO2, CT also changes. That sentence in the abstract should be re-written to accurately reflect the experiments.

The CT portion of the sentence was removed from the abstract.

2) Line 117: 20-lb

This edit was completed.

3) lines 133 and 134: see comment number 1 above.

CT was changed to PCO2 when describing the manipulated variable.

4) Lines 146 and 156: 200-mL

This edit was completed.

5) Lines 147-151: This sentence is awkward. Maybe break it up into 2 or three sentences?

This sentence was broken into two sentences.

6) Line 152: This sentence is awkward. Maybe explicitly state this is a different experiment to clarify this for the reader.

This sentence was split and states that larval experiment was started.

7) Line 153: 12-L

This edit was completed.

8) Line 167: 20-mL

This edit was completed.

9) Line 182: I suggest changing to ... 'into a 6-well plate labeled with its corresponding treatment.'

This edit was completed.

10) Please give citation for ImageJ

A citation for ImageJ was added.

11) Lines 189-194. This paragraph is a bit difficult to follow. I suggest re-writing it.

This paragraph was edited for clarity.

12) Lines 202-205 can be deleted as they appear on lines 220-222. Add the WA DFW info to the approval statement, too.

This edit was completed.

13) In the statistical analysis section of the methods, please state which stats software was used.

Added in a sentence stating R software was used and added in a citation for R. 

14) Lines 225-239 and anywhere else: Please see comment number one above.

This edit was completed.

15) Table 3: Please correct the p for pCO2. It now reads 0.0.5418

This edit was completed.

16) A recent paper by Murray and Klinger (J Exp Biol (2022) 225 (5): jeb243501) is applicable to your study and could be a good addition to your discussion.

This paper was read and added to our discussion when relevant.

---

## [Editor Report · Decision Letter 3]

13 Jun 2022

Surf smelt accelerate usage of endogenous energy reserves under climate change

PONE-D-21-27467R3

Dear Dr. Russell,

We’re pleased to inform you that your manuscript has been judged scientifically suitable for publication and will be formally accepted for publication once it meets all outstanding technical requirements.

Kind regards,

Erik V. Thuesen, Ph.D.

Academic Editor

PLOS ONE
---

## [Editor Report · Acceptance letter]

16 Jun 2022

PONE-D-21-27467R3 

Surf smelt accelerate usage of endogenous energy reserves under climate change 

Dear Dr. Russell:

I'm pleased to inform you that your manuscript has been deemed suitable for publication in PLOS ONE. Congratulations! Your manuscript is now with our production department. 

Kind regards, 

on behalf of

Dr. Erik V. Thuesen 

Academic Editor

PLOS ONE